# Modeling prominence constraints for German pronouns as weighted retrieval cues

## Abstract

We propose a cue-based retrieval model of German personal and demonstrative pronouns. The model extends the existing cue-based retrieval model of pronoun resolution by adding prominence constraints as weighted retrieval cues. We used data from an antecedent selection task reported in Schumacher et al. (2016). The experiment varied word order (canonical vs. non-canonical) and verb types (active accusative vs. dative experiencer) to test the effect of varying referential prominence on antecedent preferences for personal and demonstrative pronouns. Effectively, the model demonstrates that pronoun-antecedent dependency can be modeled as a cue-based retrieval process and the contrastive antecedent preferences of personal and demonstrative pronouns can be captured using weighted retrieval cues.

## 1 Pronoun resolution

In a sentence such as *Peter wanted to go jogging with Paula, but he had a cold*, the task of finding out what the pronoun *he* refers to involves: (i) using the linguistic knowledge that the referent should prototypically have a masculine gender, (ii) maintaining the memory representation of all the referents encountered so far, i.e. *Peter* and *Paula*, and (iii) carrying out the computation of retrieving the correct antecedent, *Peter*, and identifying it with the personal pronoun *he*.

### 1.1 Personal vs. demonstrative pronouns in German

In German, apart from the personal pronouns (PPros, henceforth) *sie/er/es* (she/he/it), there are also demonstrative pronouns (DPros) *die/der/das* (she/he/it) which are used very productively. PPros and DPros differ in their antecedent preferences. In the (1) the PPro *er* can refer to both the subject (*the firefighter*) and the object (*the boy*), but has a mild preference towards the subject antecedent.

The DPro *der* on the other hand shows a strong preference towards the object antecedent.

(1)   [Der Feuerwehrmann]$_i$ will [den Jungen]$_j$ retten, aber er$_{\{i, j\}}$/der$_{\{?i, j\}}$ ist zu aufgeregt.
[The firefighter]$_i$ wants [the boy]$_j$ to-rescue, but he$_{PPro\{i, j\}}$ / he$_{DPro\{?i, j\}}$ is too nervous
'The firefighter wants to rescue the boy, but he is too nervous.'

In general, it has been claimed that PPros prefer, whereas DPros disprefer the most salient or prominent referent (Bosch et al., 2003). Here, prominence is computed in terms of subjecthood (Bosch et al., 2007; Kaiser, 2011), agenthood (Schumacher et al., 2016, 2017), order of mention (Schumacher et al., 2016; Bader and Portele, 2019), topicality (Bosch and Umbach, 2007; Hinterwimmer, 2015), perspective taking (Hinterwimmer and Bosch, 2018; Hinterwimmer et al., 2020), or a combination of more than one of these factors (Schumacher et al., 2015; Portele and Bader, 2016 among others).

In (1) the factors of subjecthood and agenthood align such that *the firefighter* is the subject and agent of the sentence, whereas *the boy* is the object and patient of the sentence. However, when subjecthood and agenthood don't align, German pronouns show a mixed effect of subjecthood and agenthood (Schumacher et al., 2016; Patterson and Schumacher, 2021).

## 2 Data: Schumacher et al. (2016) Expt.1

Schumacher et al. (2016) carried out a set of offline studies to tease apart the effect of the factors of subjecthood, agenthood and the order of mention for German PPros and DPros. In Experiment 1, they used experimental items as in (2) where they

varied the verb type — active accusative (2a and 2b) vs. dative experiencer (2c and 2d) — and the word order — canonical (2a and 2c) vs. non-canonical (2b and 2d). Each of these four conditions occurred in two variations such that the pronoun was either a PPro or a DPro. This lead to eight conditions in total.

Table 1: Thematic and grammatical roles of the two referents across four conditions (see (2) for details of the conditions). Ref. = referent; Th. role = thematic role; Gr. role = grammatical role; AGT = agent; PAT = patient; SUB = subject; OBJ = object.

| Condition | Referent | Th. role | Gr. role |
|---|---|---|---|
| a. AA-CA | Ref. 1 | AGT | SUB |
| | Ref. 2 | PAT | OBJ |
| b. AA-NC | Ref. 1 | PAT | OBJ |
| | Ref. 2 | AGT | SUB |
| c. DE-CA | Ref. 1 | AGT | OBJ |
| | Ref. 2 | PAT | SUB |
| d. DE-NC | Ref. 1 | PAT | SUB |
| | Ref. 2 | AGT | OBJ |

(2)    a.   *Active accusative verb in canonical word order* [AA-CA]
Der Feuerwehrmann will den Jungen retten, weil das Haus brennt. Aber **er/der** ist zu aufgeregt.
The firefighter wants to rescue the boy, because the house is on fire. But **he**$_{PPro}$/**he**$_{DPro}$ is too nervous.

      b.   *Active accusative verb in non-canonical word order* [AA-NC]
Den Jungen will der Feuerwehrmann retten, weil das Haus brennt. Aber **er/der** ist zu aufgeregt.
It is the boy who the firefighter wants to rescue, because the house is on fire. But **he**$_{PPro}$/**he**$_{DPro}$ is too nervous.

      c.   *Dative experiencer verb in canonical word order* [DE-CA]
Dem Zuschauer ist der Terrorist aufgefallen, und zwar nahe der Absperrung. Aber **er/der** will eigentlich nur die Feier sehen.
The spectator has noticed the terrorist, in fact next to the barrier. But **he**$_{PPro}$/**he**$_{DPro}$ actually only wants to watch the ceremony.

      d.   *Dative experiencer verb in non-canonical word order* [DE-NC]
Der Terrorist ist dem Zuschauer aufgefallen, und zwar nahe der Absperrung. Aber **er/der** will eigentlich nur die Feier sehen.
It is the terrorist who the spectator noticed, in fact next to the barrier. But **he**$_{PPro}$/**he**$_{DPro}$ actually only wants to watch the ceremony.

This design made sure that prominence cues are not always aligned for the two referents in the first sentence. In condition (a) the first-mentioned referent (*the firefighter*) has AGENT as the thematic role and SUBJECT as the grammatical role because the verb 'retten' (*to rescue*) is an active accusative verb with a canonical nominative-accusative order. On the other hand, in condition (c) the first-mentioned referent (*the spectator*) has AGENT as the thematic role, but OBJECT as the grammatical role since the verb 'auf(ge)fallen' (*to notice*) is a dative experiencer verb with a canonical dative-nominative order. Table 1 lists the thematic and grammatical roles of the two referents across conditions (a-d). Note that the authors followed the proto-role account of Dowty (1991).

In the experiment, participants saw sentences as in (2) and performed a two-alternative forced choice task where they indicated which of the two referents in the previous sentence they preferred as the antecedent of the pronoun. Antecedent preferences across eight conditions in terms of mean percentages of choosing the first referent listed in Table 2 in the column 'Data'. The percentages for selecting the second referent are complementary percentages since it was a two-alternative forced choice task.

In sum, three important results emerged: *[Effect-1]* for active accusative verbs, where subjecthood and agenthood align, PPros preferred the referent that was subject and agent, whereas DPros preferred the referent that was object (non-subject) and patient (non-agent), *[Effect-2]* for dative experiencer verbs, the preferences were less straightforward such that in the canonical word order, the PPros preferred the first-mentioned antecedent that was object and agent, whereas DPros preferred the last-mentioned antecedent that was subject and patient; however, *[Effect-3]* in the non-canonical

condition, there was no preference for the first- or last-mentioned referent for either of the pronouns. Schumacher et al. (2016) interpreted these results as providing evidence for the interaction of multiple prominence factors, and agenthood being ranked higher than other constraints for the interpretation of PPros and DPros.

Table 2: Data and model predictions for the antecedent selection task in Schumacher et al. (2016) Experiment 1. Each cell represents the percentage of selecting the first referent (Ref. 1) in that condition. The percentages for selecting the second referent (Ref. 2) are complementary percentages since it is a two-alternative forced choice task. The first four rows are for PPros and the last four are for DPros.

|  | Condition | Data | Model 1 | Model 2 | Model 3a/3b |
|---|---|---|---|---|---|
| PPro | a. AA-CA | 62% | 41% | 82% | 86% |
| | b. AA-NC | 43% | 42% | 26% | 24% |
| | c. DE-CA | 59% | 42% | 55% | 63% |
| | d. DE-NC | 47% | 40% | 56% | 48% |
| DPro | a. AA-CA | 23% | - | - | 8% |
| | b. AA-NC | 67% | - | - | 60% |
| | c. DE-CA | 35% | - | - | 24% |
| | d. DE-NC | 52% | - | - | 34% |

## 3 Antecedent preference as cue-based retrieval

The cue-based retrieval theory (CBR, henceforth) proposed in Lewis and Vasishth (2005) and Lewis et al. (2006) has been successfully applied to model the memory retrieval processes involved in forming dependencies between two linguistic units such as noun-verb agreements (Wagers et al., 2009) and pronoun-antecedent dependencies (Dillon et al., 2013; Parker and Phillips, 2017; Patil et al., 2016; Patil and Lago, 2021). The CBR theory, which is implemented in the general cognitive architecture ACT-R (Anderson et al., 2004), describes sentence processing as a series of activation-based skilled memory retrievals. Lexical knowledge and current partial representation of the input (the parse) is maintained in declarative memory, and psycholinguistic processes are represented in procedural memory. Incremental sentence processing occurs through selection of procedural memory rules (parsing procedures) that retrieve declarative memory representations and operate on them to update the sentence representation.

Here our goal is to use existing CBR models of

pronoun resolution and test if they can be extended in a meaningful way to model the differences in terms of prominence constraints for pronouns in German. For doing so Expt.1 from Schumacher et al. (2016) provides a suitable dataset because it shows variations in antecedent preferences based on varying prominence features of the antecedents. Moreover, the data exemplifies the contrastive nature of the constraints for the two types of pronouns — PPros vs. DPros — used in the experiment (see Section 2 for details of the data).

## 4 Model of Schumacher et al. (2016) Expt.1

For modeling data from Schumacher et al. (2016), we carried out the following steps. First we implemented a baseline model, similar to the earlier CBR models of pronoun-antecedent dependency, which included a subset of the phi features as retrieval cues at the pronoun to retrieve the antecedent. Then we extended the model with prominence constraints and finally with weighted prominence constraints. To avoid overfitting the model, we restricted the modeling experiment to first implementing a model for the data from PPros. In general, PPros and DPros show opposite constraints for antecedents — PPros prefer a prominent referent and DPros disprefer a prominent referent. Hence, once a model for PPros is determined, the same model with contrasting retrieval cues should be able to capture the data for DPros.

A list of retrieval cues and their corresponding values used at the pronoun for all the models reported here is given in Table 3. All models assume that the referent retrieved by the retrieval process at the pronouns is the preferred antecedent for the pronoun. Model predictions are generated by running 10000 simulations for each model. All ACT-R parameters had the same values as used in Lewis and Vasishth (2005) except for cue-weighting in Model 3.

### 4.1 Model 1: Baseline model

The baseline model assumed that the antecedent for the PPros is retrieved using the cues 'gender' (= masculine), 'number' (= singular) and 'category' (=DP, a determiner phrase). We consider this to be a baseline model because the specification of retrieval cues was the same as the earlier CBR models of antecedent retrieval (e.g. Patil and Lago, 2021) and it did not have any extension to consider the

manipulation of prominence factors in the design of the experiment from Schumacher et al. (2016). The predictions of the model, in terms of the antecedent preferences, are shown in Table 2 in the column for Model 1. The antecedent preferences of the model are determined by calculating proportions of referents retrieved across all simulations. The model showed unanimous preference for the second referent and the preference was equal across four conditions. Clearly the model does not capture either of the three effects observed for PPros in the data (see Section 2 for the list of effects).

## 4.2 Model 2: Model with prominence constraints

We extended the baseline model by adding retrieval cues that reflected factors influencing prominence of the two referents. Effectively we added the cues for thematic roles, grammatical roles and order of mention (see Table 3). The predictions of the model, in terms of the antecedent preferences, are shown in Table 2 in the column for Model 2. The model captured *Effect-1* and *Effect-2* for the PPro: for active accusative verbs the PPro prefers the referent that was subject and agent (independent from canonicity) and for the canonical condition in the dative experiencer verbs the PPro prefers the first-mentioned referent that was object and agent. However, the model doesn't capture *Effect-3*: for the non-canonical condition in the dative experiencer verbs it predicts a preference for the first-mentioned referent that was subject and patient whereas in the data there is no clear preference for either of the two referents. This model is clearly an improvement over the baseline model since it captures data better.

## 4.3 Model 3a: Model with weighted prominence constraints

Schumacher et al. (2016) proposed that although multiple prominence-lending factors contribute to the reference resolution process, thematic role (e.g. agenthood) is a higher ranked factor among them. They suggest that the higher ranking of agenthood could be because of the general cognitive traits associated with (proto)agents because "Agents are a class of objects possessing sets of causal properties that distinguish them from other physical objects" (Leslie, 1995). We decided to add cue weighting and weight the thematic role cue higher than the other cues and test if the model performance improves. In ACT-R all retrieval cues have the same

weight, but in psycholinguistics it has been proposed that certain retrieval cues could be weighted higher than others (see for example: Parker et al., 2017; Vasishth et al., 2019; Patil and Lago, 2021).

To incorporate the importance of the thematic role cue, we modified the default *strengths of association* equation in ACT-R from Equation 1 to Equation 2 and added an extra parameter for each retrieval cue (see Anderson et al., 2004 or Lewis and Vasishth, 2005 for details about the *strengths of association* equation and its influence on the retrieval process). In Equation 2, the *CueWeight* term represents the weight of cue $j$ during the retrieval of element $i$. This modification could also be seen as modifying the value of the ACT-R parameter *maximum associative strength* for a specific retrieval cue.

$$S_{ij} = S - ln(fan_j) \qquad (1)$$

$$S_{ij} = CueWeight_j * S - ln(fan_j) \qquad (2)$$

In the modified model we weighted the retrieval cue of thematic role 1.5 times higher than other cues used to retrieve the antecedent. All other cues had a default weight of 1. Note that for cues with the default weight values, Equation 2 reduces to Equation 1, and hence the *strengths of association*, $S_{ij}$, is the same as it would be in default ACT-R; however, when the weight value is different than 1, the value for *strengths of association* reflects the weighted importance of that particular cue. The predictions of the new model are shown in Table 2 in the column for Model 3a. The modified model now also captures *Effect-3* along with *Effect-1* and *Effect-2*.

## 4.4 Model 3b: Model for DPros

In contrast to PPros, which prefer prominent antecedents, DPros are claimed to disprefer prominent antecedents. Because of this contrastive preference between the two pronouns, we predicted that the model for PPros to work for DPros with changes only in the values of the retrieval cues for prominence factors, and should not require any other changes. The corresponding modified values of the retrieval cues for DPros are listed in Table 3 in the row for Model 3b. The predictions of the model for DPros are shown in Table 2 in the column for Model 3b. The model captured *Effect-1* and *Effect-2*, however, it didn't capture *Effect-3*: for the non-canonical condition in dative experiencer verbs the model predicts a preference for the last-mentioned referent, whereas the data doesn't

Table 3: List of retrieval cues and their values across all models. The only difference in Model 2 and 3a was in terms of weighting — the cue *thematic role* was weighted to be 1.5 times higher than all other cues. Mod. = model; Cat. = (phrasal) category; Th. = thematic role; Gr. = grammatical role; Ord. = order of mention; DP = determiner phrase; M. = masculine; Sg. = singular; AGT = agent; PAT = patient; SUB = subject; OBJ = object.

| Mod. | Retrieval cue | | | | | |
|------|------|------|------|------|------|------|
|      | Cat. | Gn.  | No.  | Th.  | Gr.  | Ord. |
| 1    | DP   | M.   | Sg.  | -    | -    | -    |
| 2    | DP   | M.   | Sg.  | AGT  | SUB  | first |
| 3a   | DP   | M.   | Sg.  | AGT  | SUB  | first |
| 3b   | DP   | M.   | Sg.  | PAT  | OBJ  | last |

show any clear preference. This may indicate that DPros and PPros do not entirely show complementary interpretation preferences and are subject to form-specific weightings. This should be addressed in future research.

## 5   General discussion and conclusions

The results from the modeling experiments showed that a modified cue-based retrieval model can capture important patterns in the data for German personal and demonstrative pronouns. We started with a baseline model, in the CBR framework, for data for PPros from Experiment 1 in Schumacher et al. (2016). Since the model did not capture crucial patterns in the data that emerged due to the variations in the prominence of the referents, namely, the word order variation (canonical vs. non-canonical) and the verb type variation (active accusative vs. dative experiencer), we extended the model to include retrieval cues reflecting prominence constraints. The model that included prominence constraints performed better than the baseline model. A further improvement of the model was observed when we weighted the retrieval cues to assign a higher weight to the cue specifying the thematic role of the antecedent. Since the model for PPros with weighted retrieval cues captured all the crucial patterns in the data, we modified this model to reflect the contrast in prominence constraints between PPros and DPros, and tested its predictions for DPros. The model for DPros indeed captured two out of three crucial effects observed in the data.

In sum, the model reported here: (1) captures crucial patterns in the data from an antecedent selection task with German personal and demonstrative pronouns, (2) shows that prominence constraints on pronouns can be translated to weighted retrieval cues in the cue-based retrieval framework, and (3) shows that the contrastive antecedent preferences of personal and demonstrative pronouns can be captured with contrastive retrieval cues. We consider the model as an important step towards modeling the processing of pronouns as a cue-based retrieval process.

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
