# OpenReview forum: "Modeling prominence constraints for German pronouns as weighted retrieval cues"
_aclweb.org/ACL/2022/Workshop/CMCL — Submitted to CMCL 2022_

### Official Review · Reviewer_oGD6 · 2022-03-26
**It is difficult to find clear evidences from this short paper supporting the initial claim.**

**Rating:** 4
**Confidence:** 4

**Review:**

This paper addresses the question of pronoun reference focusing on personal and demonstrative pronouns in German. It proposes different models and study more specifically the question of prominence, showing the interest in adding this constraint in predictive model. The paper addresses an interesting question and proposes different models for studying the problem. However, it suffers from lack of details  in the description of the models, the  evaluation of the results and the discussion. It is difficult to find clear evidences from this short paper supporting the initial claim.

---

### Official Review · Reviewer_SofH · 2022-03-27
**Some critical details missing in the current submission**

**Rating:** 6
**Confidence:** 5

**Review:**


The paper describes an approach to model personal and demonstrating pronoun resolution using the ACT-R cue-based theory. The authors compare the baseline model with 3 models that differ in terms of use of prominence features (thematic role, grammatical role, order of mention) as retrieval cues and weights assigned to these cues. The results show that the models with weighted cues perform best in capturing pronoun resolution patterns found in the human data. The model for the personal pronoun resolution captures all the 3 effects, while the model for the demonstrative pronoun resolution captures 2 out of 3 effects. The weighted cues model outperforms the baseline and prominence (but no weights) models.

I found the paper to be well motivated as well as well written. However one key issue with the paper is that the claims are unsubstantiated, i.e., not enough details are provided regarding the methodology and evaluation. For example, we don't know if the difference in percentages found in the human data significantly different? Similarly, the authors claim that the models 2 and 3a/3b are able to capture various effects, e.g., "The model captured Effect-1 and Effect-2 for the PPro: ..." what is the basis of saying this? Is it based on the fact that the differences in the percentages found in the human data and the models are qualitatively similar or was there some objective way of testing this similarity? On similar lines, the authors state that "Model predictions are generated by running 10000 simulations for each model" and "The antecedent preferences of the model are determined by calculating proportions of referents retrieved across all simulations". This evaluation criterion is not backed by a rationale, for example, to make the prediction of the model more comparable with the experiment, shouldn't the model be evaluated on the same number of items that the humans were exposed to? and then to take correct proportion of PPro/DPro from them?

In sum, this is a good work, but I am unable to evaluate it with confidence given the lack of information regarding some critical issues related to methodology and evaluation.

---

### Official Review · Reviewer_ahUb · 2022-03-27
**Difficult to confirm the conclusions since key details of the model setup and evaluation are missing**

**Rating:** 4
**Confidence:** 3

**Review:**

This paper proposes a cue-based retrieval model for a really interesting pronoun resolution phenomenon in German: The distinction in reference preferences between personal vs. demonstrative pronouns, depending on different aspects contributing to a referent’s saliency. Contributing factors to a referent’s saliency that are considered include subjecthood, agenthood and order of mention, which are fleshed out in terms of (weighted) cue prominence in the new models proposed. The authors show that a model with these additional weighted prominence cues is able to account for key empirical patterns in human data found by Schumacher et al. (2006), as defined in Effects 1-3 in the paper.

**General**

The paper is well-written and provides a great overview of the phenomenon of interest and motivation for the modeling work that is its main contribution. The models seem (mostly) straightforward; however, key information about the model and evaluation methodology and are omitted, which leave the conclusions the authors want to draw unwarranted (for now).

**Specific questions/Comments**

* A key component missing in the paper is a well-defined model evaluation metric. For example, looking at Table 2, we see that Models 2 & 3a severely overestimate condition a., underestimate b., and are about in the right ballpark for conditions c. & d. Yet, Model 3a is claimed to capture the relevant variance in the human data. What are the variances for the conditions in the human data? What is the variance in the model results? Do they align? The way it reads now is a qualitative comparison of proportions where an Effect is captured if the target referent is chosen more than 50% of the times, disregarding differences in preferences between conditions in the human data, which seems unsatisfactory.
* Relatedly, is there a particular reason why there is no effect described that relates to the AA-NC condition? The baseline model seems to be closest, quantitatively & qualitatively speaking, to the human data here, whereas both other models underestimate the proportion. So *is* the baseline model capturing some crucial variance in the data that the other models are not?

* Again relatedly, is a unanimous preference for the second referent expected for the baseline model given that the retrieval cues are always consistent with both referents (provided I understand this correctly)? Shouldn’t the model perform at chance? Clarifying the model description and training regime would be helpful for this I believe.
* “In the modified model we weighted the retrieval cue of thematic role 1.5 times higher than other cues used to retrieve the antecedent”. Is it true that the thematic role cue is weighted 1.5 times higher than the baseline model cues as well, or only higher than other prominence-inducing cues (grammatical role/ order of mention)? The former seems counter-intuitive as grammatical features should provide a stronger bias for pronoun resolution than thematic roles. If my interpretation is correct, it would be good to add empirical evidence for this effect, as I don’t think this is concluded by Schumacher et al. (2016), which is their reference.

**Thoughts regarding future extensions to the modeling work (do not need to be included in the paper!)**

* As it stands the paper assumes a complementary distribution between personal pronouns *er/sie/es* and demonstrative pronouns *der/die/das*. I wonder how other demonstrative pronouns like *dieser, jener, derjenige* will figure into this constellation?
* Furthermore, given the way in which the models for PPros and DPros are set up, I wonder how this approach will extend to contexts in which there are more than two referents in the context? (cf. Patterson & Schumacher 2021)

**Minor**

* For reader orientation, it would be good to spell out the critical effects in terms of predictions that clearly relate to the human data in Table 2. I.e., Effect 1: P(first referent|AA-CA, PPro) > P(second referent|AA-CA, PPro)

Overall, this paper takes a cool approach to modeling an interesting phenomenon, but given the lack of detail in the model description and evaluation, the conclusions it draws do not seem sufficiently supported.

---

### Decision · Program_Chairs · 2022-03-29

Reject